# COVID-19 Vaccine Acceptance and Its Determinants among Migrants in Germany—Results of a Cross-Sectional Study

**DOI:** 10.3390/vaccines10081350

**Published:** 2022-08-18

**Authors:** Amand Führer, Latife Pacolli, Yüce Yilmaz-Aslan, Patrick Brzoska

**Affiliations:** 1Institute for Medical Epidemiology, Biometrics and Informatics (IMEBI), Interdisciplinary Center for Health Sciences, Medical School of the Martin Luther-University Halle-Wittenberg, 06112 Halle, Germany; 2Health Services Research Unit, Faculty of Health, School of Medicine, Witten/Herdecke University, 58448 Witten, Germany; 3Deptartment of Health Services Research and Nursing, Faculty of Health Sciences, Bielefeld University, 33615 Bielefeld, Germany

**Keywords:** COVID-19, migration background, vaccine acceptance

## Abstract

Vaccinations are a core element of infection control. Migrants have been reported to have low vaccination rates for many infectious diseases, including COVID-19. Still, determinants of migrants’ uptake of COVID-19 vaccinations are not sufficiently clear. The present study addresses this gap and examines the respective influence of three potential determinants: barriers to access, attitude towards vaccinations in general, and towards COVID-19 vaccines. The study uses a cross-sectional online survey among migrants in Germany. The questionnaire assessed the aforementioned determinants using standardized tools. Information on 204 individuals was available. The vaccination rate in the sample was 80%. Vaccinated as compared to unvaccinated respondents reported more often the absence of financial barriers (71% (95%CI: 64–73%) vs. 45% (95%CI: 28–63%)), short waiting times (51% (95%CI: 43–59%) vs. 22% (95%CI: 5–38%)), and the presence of a vaccination center close-by (91.5% (95%CI: 87–96%) vs. 69.7% (95%CI: 54–85%)). Concerning COVID-19 vaccine acceptance, the majority of respondents (68%) agreed that the vaccine is important. Unvaccinated respondents more often feared side effects, were convinced that the vaccine is not safe, and assumed that COVID-19 is not dangerous. Correspondingly, acceptance of vaccinations in general was higher among vaccinated respondents. In line with findings from previous studies, our survey found that all three determinants seem to influence migrants’ vaccination status while their overall vaccination rate was comparable to the general population. Hence, migration background per se does not sufficiently explain vaccine acceptance and further research is needed to identify subgroups of migrants that should be specifically addressed to increase their vaccination rate.

## 1. Introduction

Vaccinations make a decisive contribution to infection control and represent a key strategy for the long-term management of the COVID-19 pandemic [1]. Still, not all population groups are equally well reached by vaccination campaigns, and consequently, vaccination rates vary greatly between different population groups. Particularly, migrants have been reported to have lower vaccination rates for many infectious diseases—including COVID-19—than non-migrants [2,3,4].

The vaccination rate of a population group depends on the accessibility of the vaccination and the willingness of individuals to be vaccinated [5]. Differences in vaccine acceptance and attitudes towards vaccination may therefore contribute to disparities in vaccination rates. Similarly, access to the health system and especially to preventative health care services differs across population groups and can lead to disparities in vaccination rates, even in countries such as Germany, where vaccinations are generally free of charge.

With respect to migrants, previous studies on vaccination coverage in Europe have shown that migrants often face legal and administrative obstacles in accessing vaccinations, encounter racial and economic discrimination, and are confronted with language-related barriers [6], leading to a relatively low uptake of routine vaccinations [2]. In Germany, for instance, many health care providers are not aware of the fact that vaccinations are part of the health services available to asylum seekers according to the Asylum Seekers’ Benefits Act, and thus do not inform asylum seekers that they can access vaccinations free of charge [7]. In addition, migrants often do not receive adequate information on available vaccines through public health authorities and, therefore, often rely on word of mouth and social media as the main sources of information [8,9]. This makes them vulnerable to false information and conspiracy myths [10]. Inadequate information may also lead to unfavorable illness beliefs and problematic causal attributions [8,11]. As a result, it has been reported that attitudes critical to vaccination are more common in certain groups of migrants compared to the non-migrant population in various European countries [6]. In terms of COVID-19, for example, there were insufficiently addressed beliefs that the COVID-19 vaccination was not ‘halal’ and therefore not permitted by Islam [12].

In the context of the current pandemic, these problems gained momentum and are now widely discussed in lay [13] and scientific media [14]. Despite this increased attention to migrants’ vaccination status, it is still not sufficiently clear which determinants influence migrants’ uptake of COVID-19 vaccinations. Hereby, especially the role of barriers to accessing vaccination on the one hand, and attitudes towards vaccination on the other hand are relevant subjects. The present study addresses this gap and examines factors associated with COVID-19 vaccination. In particular, we focus on three potential determinants: barriers to access, general attitude towards vaccinations, and attitude towards COVID-19 vaccines (while considering a number of sociodemographic factors).

## 2. Materials and Methods

### 2.1. Sampling and Study Population

This cross-sectional online survey aimed to study migrants residing in Germany using a convenience sample. Inclusion criteria for respondents’ participation were: age above 18 years, migration background, and regular residence in Germany. For purposes of recruitment, the survey was advertised through social media and various online groups tailored to the respective communities. In addition, we distributed leaflets in shelters for asylum seekers using the material in those languages in which the questionnaire was available as well: Arabic, Persian, English, Spanish, and Turkish. In addition, all information was also available in the German language. Recruitment took place from September 2021 to January 2022.

### 2.2. Questionnaire

The questionnaire was available in German and five more languages (Arabic, Persian, English, Spanish, and Turkish). Initially, the questionnaire was developed in German, then it was translated to the respective languages, and pretested in each language by native speakers, following standard guidelines [15].

The questionnaire consisted of five parts: The first part inquired if respondents had experienced a SARS-CoV-2 infection, how severe their illness was, and if they were vaccinated at the time of the survey. The second part consisted of seven questions concerning barriers towards the COVID-19 vaccination, such as the accessibility of vaccination centers, waiting times, or economic barriers. Those questions were developed building on WHO’s Working Group on Vaccine Hesitancy inventory [16]. The third part consisted of nine questions based on the tool used by Neumann-Böhme et al. [17], which probes a number of attitudes towards the COVID-19 vaccines. For each question, respondents reply on a Likert scale ranging from 1 (strongly disagree) to 7 (strongly agree). For analysis, we grouped answers into ‘agreement’, ‘disagreement’, and ‘uncertain’.

The fourth part of the questionnaire consisted of a standard tool for the assessment of attitudes towards vaccination in general [18,19]. This instrument consists of ten items, each scored on a Likert scale between 1 (strongly disagree) and 7 (strongly agree). As a measure of overall vaccine acceptance, we calculated the score as the sum of the items divided by 10 [18]. The score can have values between 1 and 7, whereby higher values represent a more positive attitude towards vaccination [18]. In addition, we collected sociodemographic information (age, sex, years of education, country of birth, mother tongue, duration of stay in Germany, type of residence permit, and type of accommodation).

The English questionnaire is shown in Appendix A.

### 2.3. Statistical Analysis

For a description of the study population, its vaccination rate, reported barriers to vaccination, and the respondents’ attitudes towards the COVID-19 vaccine and vaccination in general, we performed descriptive statistics. We report absolute and relative frequencies. For general vaccination acceptance, we report the mean score and its standard deviation.

To explore the association between vaccination status and the determinants *barriers to vaccination*, *attitude towards COVID-19 vaccine*, and *attitude towards vaccination in general*, we conducted stratified analyses where we compared vaccinated and unvaccinated respondents’ results for the three determinants. Hereby we report relative frequencies and their confidence intervals, and the mean score and its confidence interval, respectively.

## 3. Results

### 3.1. Sociodemographic Characteristics of the Study Population

Overall, 204 individuals participated in the survey and completed the entire questionnaire. Of those, 108 (57%) were female. The mean age was 37 years (min: 19 years, max: 73 years) and the median duration of stay in Germany was 6.5 years, ranging from 0 to 50 years. Further, 61 respondents (32%) were born in Germany; other common countries of birth were Turkey (n = 46, 24%), Syria (n = 17, 8.9%), and Venezuela (n = 10, 5%). Most respondents (n = 154, 80%) reported a formal education of more than ten years. The most commonly reported residence status was German citizenship (n = 68, 36%), followed by permanent residence permit (n = 57, 30%) and temporary residence permit (n = 32, 17%). Further sociodemographic details are presented in Table 1.

### 3.2. Vaccination Status concerning COVID-19

At the time of the survey, 24% (n = 47) of all respondents reported having already had COVID-19, most of them (55%, n = 26) with flu-like symptoms. Further, 28% (n = 14) reported symptoms comparable to common cold, 8% (n = 4) reported no symptoms, while 4% (n = 2) were hospitalized, with one person (2%) needing mechanical ventilation. The overwhelming majority of respondents reported COVID-19 cases in their family or among friends (77%, n = 149).

In total, 80% (n = 154) of all respondents had already received at least one dose of some COVID-19 vaccine; 2% (n = 4) reported not being vaccinated but already having a vaccination appointment, while 17% (n = 33) reported not having an appointment yet. Among the subgroup of respondents who had not already have COVID-19, the vaccination rate was slightly higher than in the overall study population (85.4% vaccinated vs. 2.1% not vaccinated but already got an appointment and 12.5% not vaccinated).

### 3.3. Access to Vaccinations

Apart from COVID-19, 46% (n = 87) of respondents reported that they received a vaccination in the last five years, and also concerning the accessibility of COVID-19-related vaccination services, the overall accessibility was deemed good: The majority of respondents (87%, n = 164) reported that there is a vaccination center close by and that they could reach it easily (90%, n = 172); that it was easy for them to make an appointment (67%, n = 128); that the waiting time was not long (45%, n = 86). In addition, they mostly agreed that there were no monetary barriers for them (66%, n = 125), that their insurance status did not complicate their access to vaccination (75%, n = 142), and that language barriers did not affect their access to vaccinations (69%, n = 132).

There were some differences in the above-mentioned measures of accessibility between vaccinated and unvaccinated respondents, suggesting that accessibility might influence vaccination take-up. These differences are particularly pronounced for the question if a vaccination center is close by (agreement of 91.5% (95%CI: 87–96%) among vaccinated respondents versus 69.7% (95%CI: 54–85%) among unvaccinated respondents who are not planning to be vaccinated); if the waiting time is short (51% (95%CI: 43–59%) vs. 22% (95%CI: 5–38%)); if there are no financial barriers (71% (95%CI: 64–73%) vs. 45% (95%CI: 28–63%)). For the other outcomes, confidence intervals widely overlap, so differences might also be due to chance. More details are shown in Table 2.

### 3.4. COVID-19 Vaccination Acceptance

Asked about their overall assessment of the COVID-19 vaccine, 68.6% (n = 131) of all respondents agreed that the vaccine is important, 18.9% (n = 36) disagreed, and 7.3% (n = 14) were undecided. Correspondingly, 15% (n = 29) reported general distrust against vaccinations, while 74% (n = 142) disagreed with general distrust against vaccinations.

Asked about more details of their attitude towards the vaccine, 55% (n = 92) stated that they feared the side effects of the vaccination, while 26.7% (n = 57) did not report such worries; 47.9% (n = 92) thought that the vaccination was not safe, compared to 31% (n = 60) who thought it safe; 42.7% (n = 82) assumed that they would not benefit (vs. 37%, n = 71 who thought they would benefit) from vaccination against COVID-19 since they deemed the disease not dangerous for themselves. Still, only 26% (n = 49) agreed that “nature should best be left its course” (compared to 54% (n = 103) who disagree); 52% (n = 99) reported to believe in the effectiveness of natural remedies and home remedies and 21% (n = 40) reported fear of syringes (compared to 70%, n = 133). The overwhelming majority of respondents (92% (n = 176)) disagreed with the statement that religious reasons speak against vaccination.

As could be expected, there are some differences in the acceptance of the COVID-19 vaccine between vaccinated and unvaccinated respondents: Hereby, differences were particularly pronounced in the overall assessment of the vaccination’s usefulness (79.1% (95%CI: 72.6–85.5%) vs. 24.2% (95%CI: 7.2–41.2%)), the fear of side effects (48.7% (95%CI: 40.8–56.6%) vs. 81.8% (95%CI: 68.7–95.0%)), the conviction that the vaccine is not safe (21.4% (95%CI: 13.6–29.3%) vs. 75.8% (95%CI: 61.1–90.4%)), and the assumption that COVID-19 would not be dangerous for the respondent (31.2% (95%CI: 23.3–39.1%) vs. 63.6% (95%CI: 47.2–80.1%)). Further details are displayed in Table 2.

### 3.5. General Vaccination Acceptance

Beyond the COVID-19 vaccination acceptance, the general vaccination acceptance among respondents showed a mean of 5.07, a median of 5.35, and a range of 1.4 to 7.0. As expected, mean scores are, on average, higher among vaccinated respondents compared to unvaccinated respondents (5.4 (95%CI: 5.06–5.75) vs. 3.6 (95%CI: 2.75–4.4)). Due to overlapping confidence intervals, we do not see substantial differences according to residence status, type of accommodation, sex, education, or country of birth.

## 4. Discussion

Vaccination can be considered the most important measure for the long-term management of the COVID-19 pandemic [1]. While in the majority of high-income countries, vaccination rates in the population increased steeply in the first few months after vaccines became available—with demand often surpassing supply—soon rates leveled off at about 70–85 percent at the population level [20]; therefore, in order to increase overall vaccination rates, also groups of the population must be approached which are hard-to-reach by common vaccination campaigns. In many studies, migrants and ethnic minorities have been identified to be such hard-to-reach population groups when it comes to vaccination uptake [4,6,21]. Insights into perceived barriers and attitudes towards COVID-19 vaccines could therefore contribute to devising appropriate strategies in order to increase vaccine acceptance among this population group.

The present study used an online survey to investigate perceived barriers to vaccination and vaccine acceptance among migrants, including refugees and asylum seekers, in Germany. The majority of respondents reported COVID-19 cases among family members or friends. This is consistent with previous studies [22], which also suggest that the incidence of COVID-19 infections is high among migrants [23]. This could be explained by a number of reasons: Language barriers might make it potentially difficult to understand and to comply with implemented measures of infection control. In addition, poorer average economic circumstances combined with inadequate housing may result in limited possibilities to follow recommended measures, thus increasing the likelihood of infection [23,24].

When it comes to attitudes towards vaccination, the survey showed that vaccinated respondents had a higher acceptance towards the COVID-19 vaccine than unvaccinated respondents, while overall vaccine acceptance was comparable to other populations. In the present study, the most common reasons for skepticism towards the COVID-19 vaccine were respondents’ beliefs that the COVID-19 vaccine was not safe, and their concerns about side effects. These results are consistent with previous studies analyzing concerns in different population groups: In a study on vaccination acceptance among nurses in France, about half of the respondents (55%) reported fear of side effects [25], while in a study among health and social care workers in the UK, fear of side effects and concern about a lack of sufficient research were the most reported concerns against vaccination [26]. An online survey in the Italian general population found that about 16% of the respondents were hesitant towards COVID-19 vaccination [27], while an online survey in the Spanish general population found that 24.9% were hesitant to be vaccinated against COVID-19, and a further 26.8% would reject the vaccination [28]. In addition, a recent study in London showed that the fear of the side and long-term effects of vaccinations can result in low willingness to be vaccinated, consequently leading to poor uptake of vaccinations [29]. Respondents in that study also reported relying more on home and natural remedies than on vaccinations. As illustrated in previous research, hesitation to take up vaccination can also be attributed to fears that arise from incorrect or limited information [30,31].

Aside from respondents’ attitudes towards COVID-19 vaccination, our survey also measured respondents’ attitudes towards vaccination in general. We hereby found that scores for vaccination attitudes in our sample are comparable to the scores measured in studies in other populations: We found a mean score of 5.07, studies from the US report scores between 4.55 and 5.83 [19,32], whereas a study in the Italian general population measured a mean score of 5.48 [33].

Corresponding to this general vaccination attitude, the vaccination rate of the sample studied was 80%, around four percentage points higher than in the general population in Germany at the time of the survey [34]. Although our sample cannot be considered representative of migrants in Germany in terms of socioeconomic profile, as a comparison with census data illustrates [35], this shows that migrants are not per se a population group that is vulnerable with respect to a low vaccination rate. Our sample is characterized by, on average, high education, secure residence status, and relatively long durations of stay in Germany; therefore, it seems not to be a hard-to-reach population in the above-mentioned sense.

Still, the findings also highlight that unvaccinated respondents tended to be more likely to report barriers than vaccinated respondents. To reduce those barriers in the access to vaccination, the literature suggests that vaccination centers or physicians could provide vaccinations on different days of the week, including weekends, and at different times to increase accessibility for those with less flexible working hours. In addition, geographical accessibility using public transportation should be ensured [36]. Vaccinations can further be administered in a broad range of settings, including community centers or religious institutions. Furthermore, the possibilities of mobile vaccination clinics should be explored. For example, Loma-Linda University, California, United States, has been offering vaccinations in a church parking lot also involving faith leaders and health care professionals from the local African-American community, and was able to improve vaccination coverage considerably [37].

### Strengths and Limitations

To the best of the authors’ knowledge, this is the first study to focus on barriers to access, general attitudes towards vaccinations and attitudes towards COVID-19 vaccines as potential determinants of COVID-19 vaccination uptake among migrants that utilizes a multi-language questionnaire and a broad recruitment strategy, also distinguishing between vaccinated and unvaccinated respondents; however, some limitations of the study have to be mentioned. Because invitations to participate in the study were broadly distributed, it is not possible to calculate the actual response rate. Given the mode of administration and the selection bias associated with online surveys, our findings may not be representative of the overall migrant populations residing in Germany [38]. In particular, we observed a large number of respondents with a high educational status and a secure residence status who had lived in Germany for many years. This could also explain why the vaccination rate in our study was high. Similarly, our sample might not be representative of the whole population of migrants in Germany when it comes to other important determinants of vaccination uptake, such as trust in institutions and health literacy [19,27] or pre-existing medical conditions [28].

## 5. Conclusions

Low vaccination acceptance and access barriers contribute to overall low immunization rates in many populations and affect the effective management of the COVID-19 pandemic. Accordingly, the WHO considers vaccination hesitancy as one of the top 10 risks to global health [39]. The vaccination rate in the present study is high, which might be explained by the high education and secure residence status of the respondents. It shows that migrants are not vulnerable to poor vaccination rates per se, and emphasizes that migrants are heterogeneous in terms of their needs and experiences in their health care system. To better understand differences in vaccine acceptance across subgroups of migrants living in Germany, more research built on bigger sample sizes is needed.

In the meantime, information campaigns or measures aiming to increase vaccination readiness or uptake must take that heterogeneity into account and should focus on migrants who live in a non-permanent residence, have a low education or poor German-language proficiency, and therefore are presumably more likely to have a low vaccination rate.

## Figures and Tables

**Table 1 vaccines-10-01350-t001:** Sociodemographic characteristics of the study population.

	n	%
Sex		
Female	108	57.1
Male	81	42.9
Age [years]		
<=20	6	3.3
21–25	26	14.4
26–30	32	17.7
31–35	17	9.4
36–40	28	15.5
41–45	27	14.9
46–50	21	11.6
51–55	14	7.7
56–60	6	3.3
61+	4	2.2
Country of birth		
Germany	61	33.2
Turkey	46	25.0
Syria	17	9.2
Venezuela	10	5.4
Iran	8	4.4
Other	42	22.8
Mother tongue		
Turkish	94	49.5
Arabic	28	14.7
Spanish	27	14.2
No information	14	7.4
German	12	6.3
Farsi/Dari	11	5.8
English	3	1.6
French	1	0.5
Education [years]		
More than 10	154	80.2
10	15	7.8
Less than 10	19	9.9
No formal education	4	2.1
Duration of stay in Germany [years]		
up to 2	25	22.9
3 to 5	16	14.7
6 to 10	19	17.4
More than 10	49	44.9
Residence status		
German citizenship	68	36.2
Permanent residence permit	57	30.3
Temporary residence permit	32	17.0
No information	13	6.9
Preliminary residence permit	11	5.9
Temporary suspension of deportation	7	3.7
Type of accommodation		
Rented apartment	101	52.9
Own house	52	27.2
Shared accommodation	13	6.8
Own apartment	11	5.8
Student’s hostel	7	3.7
Shelter for asylum seekers	7	3.7

**Table 2 vaccines-10-01350-t002:** Access barriers to vaccination and attitudes towards COVID-19 vaccination.

		All		Vaccinated			Unvaccinated		
		n	%	n	%	95% CI *	n	%	95% CI *
Barriers to COVID-19 vaccination	Vaccination center close by	164	86.8	139	91.5	87.0–95.9	23	69.7	54.0–85.4
	Vaccination center can be reached easily	172	90.5	144	94.1	90.4–97.9	26	78.8	64.8–92.7
	Easy to make an appointment	128	67.0	118	76.6	69.9–83.3	20	60.6	43.9–77.3
	No long waiting time	86	45.5	78	51.0	43.1–58.9	7	21.9	5.4–38.3
	No language barriers	132	69.5	114	74.0	67.1–81.0	17	53.1	35.8–70.4
	No monetary barriers	125	65.8	110	71.4	64.3–78.6	14	45.2	27.6–62.7
	No insurance problems	142	75.1	122	79.2	72.8–85.6	18	60.0	42.5–77.5
Attitudestowards COVID-19 vaccination	I think the COVID-19 vaccine is useful.	131	68.6	121	79.1	72.6–85.5	8	24.2	7.2–41.3
	I am afraid of side effects.	57	29.7	75	48.7	40.8–56.6	27	81.8	68.7–95.0
	I think that the vaccine is not safe.	60	31.3	33	21.4	13.6–29.3	25	75.8	61.1–90.4
	I don’t think that COVID-19 is dangerous for me.	71	37.0	48	31.2	23.3–39.1	21	63.6	47.2–80.1
	I am generally against vaccinations.	29	15.1	17	11.0	4.8–17.3	11	33.3	16.3–50.3
	It is best not to mess up nature.	49	25.7	36	23.5	15.8–31.3	13	39.4	22.7–56.1
	I believe in house remedies.	99	51.6	73	47.4	39.5–55.3	22	66.7	50.6–82.8
	I am afraid of syringes.	40	21.1	31	20.3	13.2–27.4	8	25.0	8.2–41.8
	I reject vaccinations because of religious reasons.	3	1.6	2	1.3	0.00–5.00	1	3.0	0.0–16.2

* 95% confidence interval.

## Data Availability

Data are available from the authors upon reasonable request.

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
