# Peer review of "COVID-19 Vaccine Acceptance and Its Determinants among Migrants in Germany—Results of a Cross-Sectional Study"

_vaccines, 2022, doi:10.3390/vaccines10081350_

Round 1
Reviewer 1 Report
Congratulations to the Authors for their manuscript, which is well written and clearly presented.
A few minor comments before it can be published:
- it would have been interesting to evaluate the level of health literacy of the sample and then assess if different levels of health literacy were associated with the willingness to get vaccinated. This may be added to the discussion/limitations as a "discussion point".
- you correctly point out that "previous studies [21], also suggest that the incidence of COVID-19 infections is higher among migrants [22]". I suggest also discussing the differences between the acceptance of COVID-19 vaccinations in migrants and the general population in European countries, measured with online surveys. It would give the reader a better context of the situation and understand the differences between the two populations (examples are 1-3)
1 https://pubmed.ncbi.nlm.nih.gov/34200656/
2. https://pubmed.ncbi.nlm.nih.gov/34063476/
3. https://pubmed.ncbi.nlm.nih.gov/33669441/
Author Response
Dear Sir or Madam,
We thank the reviewers for their helpful comments and suggestions, and the editor for the opportunity to revise our manuscript. Below, we answer to each of the reviewers’ suggestions one by one, while the changes in the manuscript are marked in the track-change mode:
- It would have been interesting to evaluate the level of health literacy of the sample and then assess if different levels of health literacy were associated with the willingness to get vaccinated. This may be added to the discussion/limitations as a "discussion point".
We agree with the reviewer that such an analysis would be interesting. Unfortunately, information on health literacy were not collected in our study. We added the reference to health literacy (and other potential confounders) in the discussion, where we discuss it in terms of the sample’s potential biases.
- You correctly point out that "previous studies [21], also suggest that the incidence of COVID-19 infections is higher among migrants [22]". I suggest also discussing the differences between the acceptance of COVID-19 vaccinations in migrants and the general population in European countries, measured with online surveys. It would give the reader a better context of the situation and understand the differences between the two populations (examples are 1-3)
We thank the reviewer for this idea. We included this topic in the introduction and added more references there. In addition, we now refer to the vaccination acceptance in different study populations in the discussion, where we compare the hesitancy in other populations to the hesitancy in our sample.
Reviewer 2 Report
The study's aim is essential and significant for vaccine research.
Please consider the following changes to improve the manuscript:
1. Please add more data (number, %, p-values) in the Abstract section - lines 23-25
2. Sampling method is unclear. Please add a more detailed description of how the participants were recruited, what type of sampling was used, do the sample provide some representativeness of the population?
3. Data collection time was relatively long. How the new wave of the COVID-19 pandemic (Autumn/Winter) may affect the data collection process?
4. I have concerns about the diversity of the group. There are multiple different migrant groups in Germany (Eastern European, Turkey, Africa, etc.). How does this study address all these groups?
5. Brief info on access to vaccines among migrants should be provided (As well as its costs).
6. Statistical analysis is quite simple. Please consider regression analysis.
7. Please add 2-3 sentences on the practical implications of this study and further research needs
Author Response
Dear Sir or Madam,
We thank the reviewers for their helpful comments and suggestions, and the editor for the opportunity to revise our manuscript. Below, we answer to each of the reviewers’ suggestions one by one, while the changes in the manuscript are marked in the track-change mode:
1. Please add more data (number, %, p-values) in the Abstract section - lines 23-25
We added the frequencies of the respective answers and their confidence intervals to the abstract.
2. Sampling method is unclear. Please add a more detailed description of how the participants were recruited, what type of sampling was used, do the sample provide some representativeness of the population?
We added more details to our description of the sampling method in the methods section. In addition, we would like to draw the reviewer’s attention to the limitations section (page 9, lines 274-280), where we discuss the limited representativeness of the collected sample.
3. Data collection time was relatively long. How the new wave of the COVID-19 pandemic (Autumn/Winter) may affect the data collection process?
We agree with the reviewer that the time for data collection has been relatively long; we had to navigate the tradeoff between allowing for a longer collection period and not reaching a sufficient sample size. Still, during the collection period there were no major changes related to laws concerning the pandemic or the availability of vaccinations in Germany. We therefore think that the longer collection period did not impair the quality of our data.
4. I have concerns about the diversity of the group. There are multiple different migrant groups in Germany (Eastern European, Turkey, Africa, etc.). How does this study address all these groups?
We addressed this issue by providing the questionnaire in multiple languages (Arabic, Persian, English, Spanish, and Turkish); We also wanted to include French, but the translation of the questionnaire into French was delayed, so we had to omit it from the study. We changed the way we describe the questionnaire in the methods section to make clear that it was available in these five languages.
In terms of online recruitment, we selected social media channels that our working group has used for recruitment in previous research and are known to be highly frequented by the different groups of migrants.
5. Brief info on access to vaccines among migrants should be provided (As well as its costs).
We added one sentence that addresses the topic of vaccination for asylum seekers and its problems, and added the information, that for all other migrants vaccinations are also available free of charge.
6. Statistical analysis is quite simple. Please consider regression analysis.
We agree with the reviewer’s opinion that a regression analysis would have been helpful to further investigate potential differences between vaccinated and unvaccinated respondents, and the risk factors for not having a vaccination.
Unfortunately, our sample size and the small size of many of the relevant subgroups severely limit the possibilities for more complex analyses.
7. Please add 2-3 sentences on the practical implications of this study and further research needs
We would like to draw the reviewer’s attention to the Conclusion of the manuscript, where we already outline practical implications. Taking up the reviewer’s suggestion, we added an explanation here, where we define further research needs.
Reviewer 3 Report
I think this is altogether a well done cross sectional study about the attitude against covid vaccination in Germany among people with backgroudn of migration. What surprises and also is nice to see is the fact that there is obviously no difference between different types of pupulation that comes with an attitude against vaccination. Why is this not more clearly stated? Is it because the authors think that the return of the survey was not waht they had expected? Would it have been more effective or would it have been better to performe interviews in certain populations to get better information?
Concerning Material and Methods I miss some kind of statistical evaluation. I think this shoud be mentioned and added. Did the authors perform any identification of subgoups or characteristics that show a differnt attitude towards vaccination- was there any difference between place of birth, mother language or education? This could be stated more clearer.
Do the authors think a control group with just a German pupulation might have made sense?
Coudl the authors publish the questionnaires in an addendum?
Author Response
Dear Sir or Madam,
We thank the reviewers for their helpful comments and suggestions, and the editor for the opportunity to revise our manuscript. Below, we answer to each of the reviewers’ suggestions one by one, while the changes in the manuscript are marked in the track-change mode:
- What surprises and also is nice to see is the fact that there is obviously no difference between different types of pupulation that comes with an attitude against vaccination. Why is this not more clearly stated? Is it because the authors think that the return of the survey was not waht they had expected? Would it have been more effective or would it have been better to performe interviews in certain populations to get better information?
We thank the reviewer for raising these important questions. Indeed, it would be interesting to better understand our respondents’ attitude towards vaccination in depth, but conducting such interviews would have been beyond the scope of our project.
That said, we have no reservations concerning the validity of our findings: Migrants in Germany are a heterogeneous group and campaigns targeting migrants should take that into account, while the relevance of migration background should always be carefully weighed against the existing evidence. Our study showed that in this well-educated sample of migrants with mostly secure residence statuses, vaccination rates are similar to those in other populations.
Since we agree with the reviewer that this is an interesting finding, we added one sentence to the abstract to highlight this conclusion early on in the article.
- Concerning Material and Methods I miss some kind of statistical evaluation. I think this shoud be mentioned and added. Did the authors perform any identification of subgoups or characteristics that show a differnt attitude towards vaccination- was there any difference between place of birth, mother language or education? This could be stated more clearer.
This suggestion aligns with the comment of reviewer 2. We agree that a more elaborate statistical analysis would have been desirable, but the sample size and the small size of many of the subgroups impose a limit on the statistical options. Thus, we see no viable way to assess the impact potential confounders like mother language or education have on the relation of vaccination status and the three determinants of interest. Still, if the reviewer is aware of any statistical approach that we might have missed, we are very grateful for further suggestions.
- Do the authors think a control group with just a German pupulation might have made sense?
In general, a control group might have made it possible to directly answer the question how migration background affects the respondents’ vaccination status. Meanwhile, this was not our research question, and would anyway have been limited by the likely limits to the representativeness of both, migrant and non-migrant study population. We therefore think that a non-migrant German control group would not have substantially improved our ability to answer our research question.
- Coudl the authors publish the questionnaires in an addendum?
We thank the reviewer for this suggestion. We added the questionnaire as a supplement to the manuscript.
Round 2
Reviewer 2 Report
The manuscript was revised in line with the comments suggested by the reviewer. The manuscript is scientifically correct and may be accepted for publication in Vaccines.
Reviewer 3 Report
no more comments, mansucript coud be published as it is